# Recent Advances in Alternative Cementitious Materials for Nuclear Waste Immobilization: A Review

## Nailia Rakhimova

Department of Building Materials, Institute of Building Technology, Engineering and Ecology Systems, Kazan State University of Architecture and Engineering, 420043 Kazan, Russia; nailia683@gmail.com;
Tel.: +7-906-323-4529

**Abstract:** Since the emergence of the problem of nuclear waste conditioning, cementation has become an important and developing part of the waste management system, owing to its simplicity and versatility. The continued development of the cementation technique is driven by the improvement and expansion of cementitious materials that are suitable and efficient for nuclear waste solidification. Advances in cement theory and technology have significantly impacted improvements in nuclear waste cementation technology, the quality of fresh and hardened waste forms, waste loading rates, and the reliability and sustainability of the nuclear industry. Modern mineral matrices for nuclear waste immobilization are a broad class of materials with diverse chemical–mineralogical compositions, high encapsulation capacities, and technological and engineering performance. These matrices include not only traditional Portland cement, but also non-Portland clinker inorganic binders. This review focuses on recent trends and achievements in the development of calcium aluminate, calcium sulfoaluminate, phosphate, magnesium silicate, and alkali-activated cements as cementitious matrices for nuclear waste stabilization/solidification.

**Keywords:** cement; nuclear waste; solidification

## 1. Introduction

Nuclear energy, which is characterized by a low carbon footprint, high power density, and ability to generate electricity quickly, holds a stable position among other forms of energy and is currently considered one of the most viable forms of base-load electrical generation for the next 50–100 years [1–3]. Recent statistics indicate that nuclear energy accounts for 10% of global power production and is anticipated to rise to 1/3 of world power by 2060 [4]. However, because one of the challenging deficiencies of nuclear power is the generation of radioactive waste (RW), further sustainable development of the nuclear industry must be accompanied by the consistent development of an RW management system.

According to the definition given by the International Atomic Energy Agency basic safety standard "Radiation protection and safety of radiation sources: International basic safety standards" [5], radioactive waste is material that contains, or is contaminated with, radionuclides at activity concentrations greater than clearance levels, as established by the regulatory body, for which, for legal and regulatory purposes, no further use is foreseen. The terms "radioactive waste" and "nuclear waste" are generically used as synonyms in the context of safety and waste management. Nuclear waste management is a complex system and comprises all administrative and operational activities involved in the handling, pre-treatment, treatment, conditioning, transport, storage, and disposal of RW [6]. The continuous improvement of each stage makes an important contribution to improving the efficiency and reliability of the entire system [7–16].

Extensive experience has been accumulated in the field of environmentally benign RW handling, and a wide range of approaches, methods, and materials for the treatment and conditioning of diverse RW have been developed and adopted over the history of

the generation and use of nuclear power. Because of the simplicity and versatility of solidification/stabilization of RW by cementitious materials, as well as long-term testing and experience gained by practical usage, this technique is one of the main and most widely used methods for converting various low- and intermediate-level radioactive wastes (LILW) into a safe form [17–19].

RW cementation remains a developing area in the RW management system. Progress in this field has also been supported by significant achievements in materials science, particularly in the chemistry of inorganic binders, as well as in the processing of a wide range of materials. Advances in cement theory and technology significantly impact improvements in RW cementation technology, the quality of fresh and hardened wasteforms, RW loading, and the reliability and sustainability of the nuclear industry. Portland cement (PC) has been the only cementitious material for LILW immobilization for a long time. The consistent implementation of supplementary cementitious materials and chemical additives for PC and PC concretes in recent decades and achievements in this area led to the increasing application of blended and modified PC-based systems in RW cementation [20–22]. It turned out the introduction of pozzolanic and chemical modifiers into PC remarkably improved not only the technological and engineering performance of waste forms but that it could also be used to control the reaction products assemblage and structure of hardened materials, action mechanism of immobilized contaminants, and mineral matrix. Consequently, PC in combination with mineral and chemical modifiers is now normally used and adopted into the practice of RW cementation.

In the past decades, non-Portland clinker alternative binders have emerged as another progressively developing direction. Non-Portland clinker alternative binders are promising for sustainable development of the cement industry and for advancing RW solidification technology [23–32]. Alternative or so-called non-traditional cements form a large group of binders, significantly differing from PC and from each other in terms of the composition and type of the raw materials, composition of the reaction products, the mechanism of formation of the hardened cement pastes, research experience, adoption, and practical application. Most alternative cements are special cements developed in attempts to eliminate the ecological and technical disadvantages of PC and/or develop special binders for special non-building applications. Certain alternative binders presented in Table 1 have become promising for the partial replacement of PC for RW solidification.

**Table 1.** Alternative cements and their characteristics.

| Cementitious Material (Abbreviation) | Reaction Products Providing RW Immobilization (Reaction Process Mechanism) | Specific Properties | Adoption Experience |
|---|---|---|---|
| Calcium aluminate cement (CAC) | $2CaO \cdot Al_2O_3 \cdot 8H_2O$, $Al(OH)_3$ (dissolution-precipitation) | fast hardening, high strength, low permeability, high freeze-thaw, corrosion resistance | France [33] |
| Calcium sulfoaluminate cement (CSAC) | $3CaO \cdot Al_2O_3 \cdot 3CaSO_4 \cdot 12H_2O$, $3CaO \cdot Al_2O_3 \cdot 3CaSO_4 \cdot 32H_2O$, $Al(OH)_3$ (dissolution-precipitation) | | |
| Magnesium silicate cement (M-S-H cement) | M-S-H gel (dissolution-precipitation) | | |
| Phosphate cements | Magnesium phosphate cement (MPC) | Fast setting, high early strength, adhesive properties, low water demand and drying shrinkage, high temperature and chemical resistance | Russian Federation, USA [34–38] |
| | Calcium phosphate cement (CPC) | | |
| Alkali-activated cement | Alkali-activated slag cement (AASC) | Fast setting, high strength, low porosity, and high temperature and chemical resistance | Ukraine [39] |
| | Geopolymer (GP) | | Australia, Czech Republic, Slovak Republic, France, USA [40–45] |

Years of experience in research and application have demonstrated the effectiveness of alternative binders for the immobilization of toxic materials and RW [23–32]. The reactive phases obtained with alternative cements have diverse compositions compared with those obtained with PC, where the former are characterized by lower solubility and better ion exchange properties, different pH, hardened pastes demonstrate faster hardening, lower permeability, and durability. Moreover, due to the complexity of the starting materials, the cement flexibility is impacted by a greater number of contributing factors, enabling versatile design of the cementitious wasteforms, control of the composition of the reaction phases, and the achievement of desirable performance. Therefore, alternative binders have expanded the possibilities and perspectives for the cementation of toxic materials and RW from the following aspects:

— In some cases, higher efficiency for both physical and chemical immobilization of heavy metals and radionuclides;
— Widening the acceptance of wastes that can be treated and conditioned by cementation;
— Optimizing waste cementation technology in cases of problematic waste components, providing faster curing of cementitious wasteforms, and eliminating the need for waste pre-treatment;
— Enabling the use of alternative binders as adsorbents and chemical additives.

The scientific and practical interest in the use of cementitious materials as a whole and the research and adoption of alternative cements, in particular, has only increased in recent years. This study reviews recent trends and achievements in the development of calcium aluminate, calcium sulfoaluminate, phosphate, magnesium silicate, and alkali-activated cements as cementitious matrices for RW stabilization/solidification.

## 2. Alternative Cements as Cementitious Materials for RW Immobilization

### 2.1. Calcium Aluminate (CAC) and Calcium Sulfoaluminate Cements (CSAC)

The production of CAC and CSAC cements, similar to PC, is based on the heat treatment of a prepared mixture of natural rocks, including limestone, bauxite, and gypsum (in the case of CSAC), leading to the formation of hydration hardening minerals. The mineral composition of the resultant clinker of CAC and CSAC cements differs from that of PC, and heat treatment of the former two cements is carried out at lower temperatures.

#### 2.1.1. CACs

CAC consists mainly of monocalcium aluminate ($CaO \cdot Al_2O_3$ (CA)), with some secondary minerals, such as $CaO \cdot 2Al_2O_3$ ($CA_2$) and $12CaO \cdot 7Al_2O_3$ ($C_{12}A_7$) [46,47]. At temperatures of 22–30 °C in the presence of water, CA is gradually converted into dicalcium aluminate hydrate ($2CaO \cdot Al_2O_3 \cdot 8H_2O$) in the form of lamellar crystals of a hexagonal system through a dissolution–precipitation mechanism. Simultaneously, gel-like aluminum hydroxide, $Al(OH)_3$, with sorptive properties is formed.

CAC is an effective matrix for immobilizing RW as it undergoes fast hardening, and the hardened CAC pastes have high strength, low permeability, and high freeze–thaw and corrosion resistance. The sorptive ability of $Al(OH)_3$ and ion-exchange ability of ettringite (formed with the introduction of lime and calcium sulphate) [48,49] provide the chemical binding of many contaminants by CAC-based matrices. It is necessary to add to the listed advantages the applicability of CAC in respect of tolerance with the waste components retarding the setting and hardening of PC. The feasibility of CAC and blended CAC for cementation of waste containing Cr, Cd, Pb, Zn, Mg, Sn, Cs, liquid borates, radioiodide, etc., has been demonstrated [50–54].

The chemical binding of hazardous and radioactive contaminants by CACs can be improved by introducing various mineral admixtures in which reaction products with ion-exchange properties are formed. It is feasible to include up to 50% calcium sulfate, in the form of gypsum or anhydrite [48] or 5–10% slaked or non-slaked lime and limestone [49], into CAC-based cement matrices. These additives form calcium sulfoaluminate and provide

siliceous mineral materials for the formation of zeolitic phases [52], thereby improving the technological and physical–mechanical properties of CACs.

CACs are used in France for (non-radioactive) hazardous waste encapsulation [33].

Recently, CAC has been studied for the solidification and stabilization of ion-exchange resins (IERs) [55,56], as well as Sr and Cl ions [57,58]. Kononenko et al. [55,56] reported that CAC introduced with $^{137}$Cs sorbent (modified diatomite) can be incorporated with 22–25% more of a mixture of ion-exchange resins (IERs) ($Na^+$, $NO^{3-}$) and 50–83% of a ($Na^+$, $B_4O_7^{2-}$) mixture, as compared with PC-based matrices. In order to prevent the decomposition of $CaO \cdot 1.64Al_2O_3$ under the action of $B_4O_7^{2-}$ and preclude the accompanying decrease in the strength of the wasteforms, the authors proposed suppressing the reactivity of $B_4O_7^{2-}$ ions by treating IERs with alkaline earth metal (Ca, Sr, Ba) nitrates, resulting in the formation of insoluble alkaline earth metal tetraborates.

### 2.1.2. CSAC

Regardless of the merits of PC, one drawback is the shrinkage of PC-based materials. Attempts to create shrinkage-free cement have led to the development of CSACs. These special cements can be shrinkage compensating, expansive, and self-stressing, and are used for various purposes. The invention and manufacture of CSACs were introduced between the 1960s–1970s of the last century. Currently, CACS are produced in industrial volumes. Since 2004, 1.2–1.3 million tons of CACs has been produced globally each year [59].

The main mineral in CSAC, comprising 30–70%, is tetracalcium trialuminate sulfate $C_4A_3\overline{S}$ ($4CaO \cdot Al_2O_3 \cdot SO_3$) ye'elimite, also known as Klein's compound [60]. The second most important mineral in sulfoaluminate–belite cements is belite $C_2S$–$2CaO \cdot SiO_2$ [61,62].

CSA clinker is ground simultaneously with 25% gypsum for the purpose of regulating the setting, strength, strength development, and soundness. Introducing other admixtures, such as PC and limestone, is also possible and effective [30,63,64]. During the interaction of calcium sulfoaluminate with water, calcium monosulfoaluminate hydrate (AFm), and aluminum hydroxide are formed as follows:

$$C_4A_3\overline{S} + 18H \rightarrow C_3A \cdot C\overline{S} \cdot 12H + 2AH_3$$

The AFm phase belongs to the lamellar double hydroxide family. Its crystal structure is composed of positively charged main layers of $[Ca_2Al(OH)_6]^+$ and negatively charged interlayers of $[1/2\,SO_4 \cdot nH_2O]^-$ [65].

In the presence of gypsum, the interaction of $C_4A_3\overline{S}$ with water is accelerated. In addition to amorphous aluminum hydroxide, ettringite ($3CaO \cdot Al_2O_3 \cdot 3CaSO_4 \cdot 32H_2O$ or $C_3A \cdot 3C\overline{S} \cdot 32H$) is formed, where the molar ratio of $C_4A_3\overline{S}$ to $C\overline{S}$ is no less than 1:2. This process occurs via the following reaction:

$$C_4A_3\overline{S} + 2\,C\overline{S}H_2 + 34H \rightarrow C_3A \cdot 3C\overline{S} \cdot 32H + 2AH_3$$

A mixture of ettringite ($C_3A \cdot 3C\overline{S} \cdot 32H$) and $C_3A \cdot C\overline{S} \cdot 12H$ may also be formed, with the full consumption of gypsum in the reaction.

Ettringite is composed of positively charged $[Ca_3Al(OH)_6]^{3+}$ columns and negatively charged channels of $[3/2SO_4 \cdot nH_2O]^{3-}$. The structural flexibility of AFm and ettringite in terms of ion exchange provides CSACs with the ability to chemically bind many elements of both anionic and cationic nature [66–68] (Figures 1 and 2).

The intensive hydration of $C_4A_3\overline{S}$ and the binding of free water enable fast consolidation of the structure, short setting, and accelerated strength development. Ettringite formation is accompanied by a volume increase in the solid phase [49]. The hydration of $C_2S$ causes prolonged strength development and facilitates the relaxation of pressure in the crystallization of ettringite. Strätlingite ($C_2ASH_8$), C-S-H, $CAH_{10}$, or siliceous hydrogarnets can also be formed depending on the clinker composition, presence, and type of supplementary cementitious materials [30].

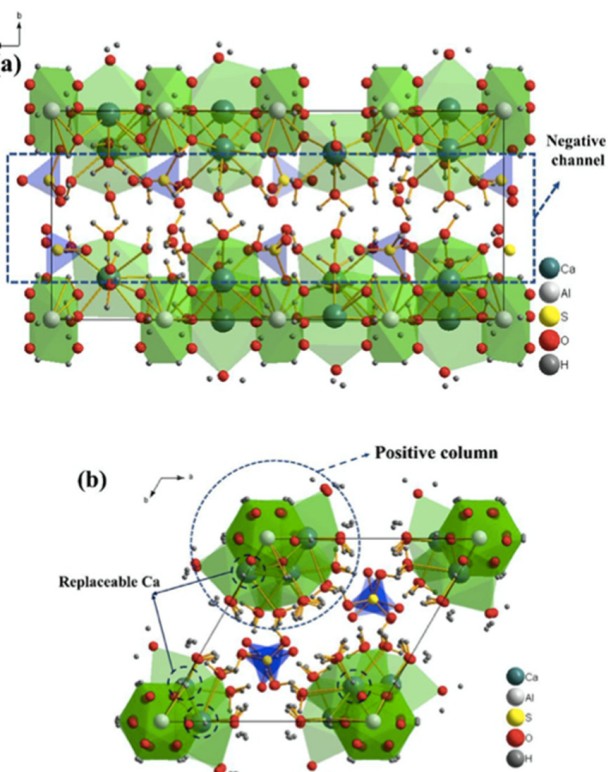

**Figure 1.** Structure of ettringite: positive columns $[Ca_3Al(OH)_6]^{3+}$ (green), negative channel $[3/2SO_4 \cdot nH_2O]^{3-}$ (blue), and replaceable Ca (green black); (**a**) c view, (**b**) b view [68].

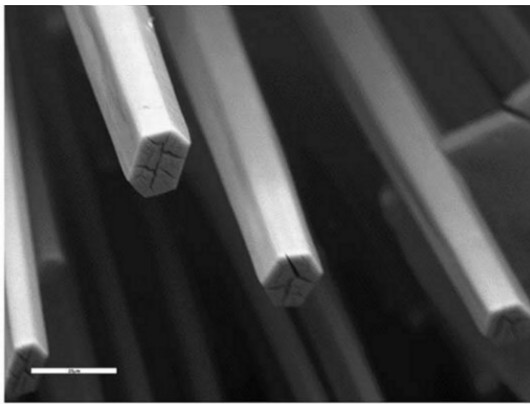

**Figure 2.** Scanning electron microscope image of ettringite crystals. Bar = 20 μm [69].

Numerous studies [30,70–74] have shown the following features of CSACs as matrices for the immobilization of toxic substances and RW:

- Accommodate heavy metals (Cr, Pb, Zn, Cd, etc.) and IERs;
- Enable the immobilization of waste that is problematic for immobilization by PC, such as those containing Al and U and wastes that produce hydrogen by interacting with cement and radioactive sludge with a high content of sulfate and borate ions;
- Allow for the precipitation of radionuclides as hydroxides (for example, $Sr(OH)_2$) due to the lower pH of hydrated CSACs while potentially decreasing the corrosion reactions of some encapsulated metals, such as Al;
- Shorten the waste cementation process owing to the high rate of hydration, and avoid pre-treatment, enabling the solidification of wastes containing components that make setting and hardening of PC-based systems difficult, for example, B, Zn, and waste.

Studies on the mechanism of solidification of borates by CSACs depending on the presence and content of gypsum and the pH have been continued by Champenois et al. [72], Chen et al. [75], and Cau-Dit-Coumes et al. [70]. Chen et al. [75] observed a dense amorphous ulexite layer with a foil-like morphology and a thickness of approximately 100 nm (Figure 3). This layer fully covered the surface of CSA clinker particles three days after mixing with 0.5 M borate solution at pH < 7, which strongly impeded the dissolution of ye'elimite. Champenois et al. [72] studied the hydration of CSAC incorporated with 0 and 10% gypsum in the presence of 1 mol/L borate ions at pH 11 and revealed that the retardation reaction of the fresh paste increased with the gypsum percentage and was correlated with the content of ulexite ($NaCaB_5O_9 \cdot 8H_2O$). The gypsum content affected the pH of the cementitious system and, consequently, the amount of ulexite formed. Cau-Dit-Coumes et al. [70] investigated the combined influence of lithium hydroxide (as an accelerator) in CSAC and sodium borate on the hydration of CSACs containing 0 or 10% gypsum. The simultaneous presence of borates and lithium led to the superimposition of acceleration and retardation effects. In the gypsum-free system, lithium promoted precipitation of the borated AFm phase. Authors believe that lithium salts can counteract the retardation caused by sodium borate. The results presented by Xu et al. [68] and Guo et al. [76] contribute to the understanding of the mechanism of immobilization of contaminant simulants in ettringite. Xu et al. [68] reported that hardened CACs due to binding capacity of ettringite and $Al(OH)_3$ along with the dense physical structure of hardened paste were better in the leaching performance of $Cs^+$ and $Sr^{2+}$, in comparison with the PC-based cementitious matrix (Figure 4). The authors proposed two superimposed mechanisms of $Cs^+$ leaching: (i) a first-order reaction between the surface of the radwaste matrix and the leachant, (ii) diffusion of $Cs^+$ through the waste matrix, (iii) release of loosely bound $Cs^+$; (i) diffusion/dissolution of $Cs^+$ and (ii) release of loosely bound $Cs^+$. The authors described $Sr^{2+}$ leaching using a combination model, including the dissolution and diffusion of $Sr^{2+}$ and the release of loosely bound $Sr^{2+}$ in the wasteform. Guo et al. [64] revealed differences in the interaction mechanisms of $I^-$, $IO^{3-}$, and ettringite. The authors observed minimal $I^-$ incorporation into ettringite (0.05%), whereas $IO^{3-}$ demonstrated high affinity for ettringite through anion substitution for $SO_4^{2-}$ (96%). Substituting $IO^{3-}$ for $SO_4^{2-}$ was energetically favorable (−0.41 eV), whereas an unfavorable substitution energy of 4.21 eV was observed for $I^-$ substitution.

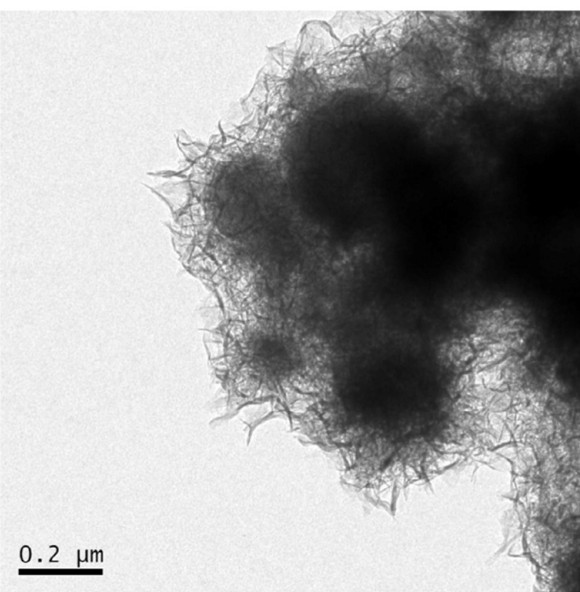

**Figure 3.** TEM image of powder CSA clinker after 3 days hydration with borate solution [75].

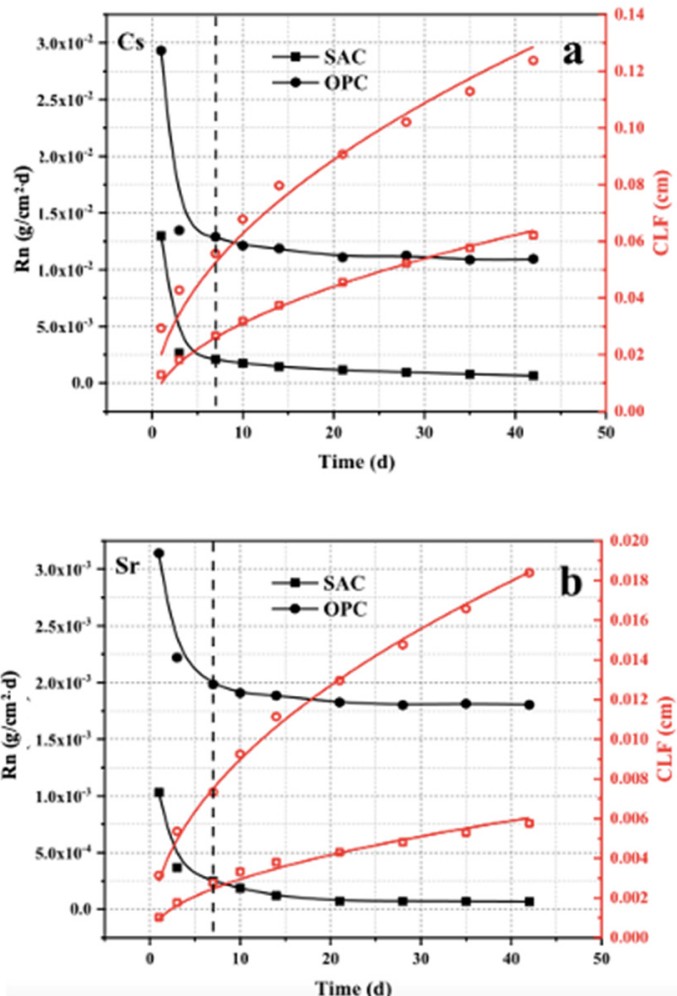

**Figure 4.** Leaching rate (Rn) and cumulative leaching fraction (CLF) fitting curves for Cs$^+$ (**a**) and Sr$^{2+}$ (**b**) [68].

Xu et al. [77] improved the performance of cementitious wasteforms based on CSAC and IERs by incorporation of MK in order to increase the resistance of IERs to prolonged water immersion. Cemented by optimal composition of 40 wt.% spent resin, 55.8 wt.% sulfoaluminate cement, 2.2 wt.% MK, and 2 wt.% water reducer, the resin loading in wasteforms was as high as 64% and the compressive strength of hardened wasteforms was 13.7 MPa. It is supposed that MK as an Al source promotes the formation of ettringite, thereby improving the stability of the solidified IERs in acidic environments or during frequent freezing-thawing. Moreover, a greater amount of ettringite provided the retention of Cs(I), with a 42 d leaching rate of $2.3 \times 10^{-4}$ cm/d.

### 2.2. Phosphate Cements

The production of phosphate binders is based on the synthesis of phosphate compounds using acid-base reactions of solids of a basic nature (CuO, FeO, ZnO, CaO, MgO, etc.) and highly reactive liquid activators comprising phosphate anions. Activators, such as aqueous phosphoric acid (mainly orthophosphoric acid $H_3PO_4$) and acid phosphate salt solutions, can be used. These include solutions of $KH_2PO_4$, $NH_4H_2PO_4$, and $CaHPO_4$. In addition to fundamental differences in the mechanism of the formation of the hardened paste, phosphate binders have a wider chemical and structural composition than other types of binders, as proven by the evaluation of a number of fundamental chemical characteristics of phosphate compounds.

Orthophosphoric acid is polybasic and has three stages of ionization, enabling multi-dimensional stereometric chemical binding and the formation of numerous connection options with varying degrees of substitution (mono-, di-, and trisubstituted salts). An important source of strength formation is also the structural characteristics of phosphoric acid and phosphates with a branched network of hydrogen bonds. Finally, orthophosphoric acid and its derivatives have a high predisposition for association with functional groups, polycondensation, and complexation [78].

The powder part of phosphate cements influences the binding properties of the systems "oxide-phosphoric acid" and the ionic potential of cations in the oxide. The conditions for activating the binding properties of the oxide-orthophosphoric acid systems are listed in Table 2.

**Table 2.** Conditions of exhibiting binding properties of oxide-orthophosphoric acid systems [47].

| Oxide | Electron Work Function, eV | Ionic Potential of Cation, z/r | Conditions of Exhibiting Binding Properties |
|---|---|---|---|
| $SiO_2$, $TiO_2$, $Al_2O_3$, $ZrO_2$, $MnO_2$, $Cr_2O_3$, $Co_2O_3$, $SnO_2$ | >4.5 | 5.0 | Intensification of acid-base interactions required |
| $Fe_2O_3$, $Mn_2O_3$, NiO, CoO, FeO, CuO | 3.3–4.3 | 2.5–4.4 | Hardening in normal conditions |
| $Nd_2O_3$, $La_2O_3$, MgO, ZnO, CdO | 2.5–3.3 | 2.0–3.0 | Passivation of acid-base interactions required |
| CaO, SrO, BaO, PbO | <2.0 | 1.4–2.0 | Emergency measures of passivation of acid-base interactions required |

Magnesium (MPCs) and calcium phosphate cements (CPCs) are the most promising for RW cementation purposes. These cements have been widely studied and have already obtained practical adoption.

2.2.1. MPCs

The raw materials for MPCs are orthophosphoric acid (or $(NH_4)_2HPO_4$ (diammonium hydrogen phosphate) and MgO, which is a product of the thermal treatment of magnesite $MgCO_3$. The main reaction product of these interactions is struvite ($NH_4MgPO_4 \cdot 6H_2O$) produced by the reaction $MgO + (NH_4)2HPO_4 + 5H_2O \rightarrow NH_4MgPO_4 \cdot 6H_2O + NH_3$, which determines the setting and hardening of this type of phosphate cement. Mixing MgO with $KH_2PO_4$ results in the formation of struvite-K.

The performance of MPCs is mainly controlled by the magnesium to phosphorus (M/P) ratio and the water/solid (W/S) ratio [79].

MPCs combine the high ion-exchange capacity of struvite, near-neutral pH, and high physical–mechanical properties of hardened paste, such as quick setting, high early strength, adhesive properties, low water demand and drying shrinkage, high temperature, and chemical resistance [34,80,81]. The struvite structure is able to take on many elements (Figure 5) [34], including monovalent cations ($NH^{4+}$, $K^+$, $Rb^+$, $Cs^+$, $Tl^+$), divalent cations ($Mg^{2+}$, $Ni^{2+}$, $Zn^{2+}$, $Co^{2+}$, $Cd^{2+}$, $Cr^{3+}$, $Mn^{2+}$, $VO^{2+}$), and trivalent oxyanions ($PO_4^{3-}$ and $AsO_4^{3-}$) [34] (Figures 5 and 6).

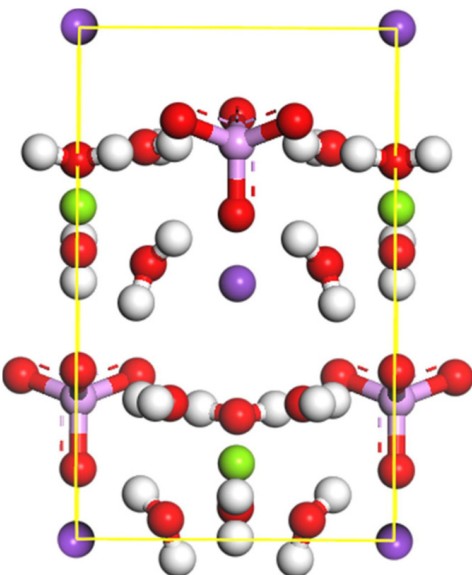

**Figure 5.** Crystal structure of struvite [82].

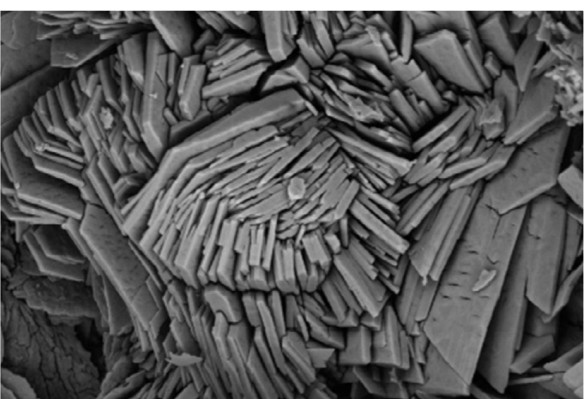

**Figure 6.** Micrographs of the MPC sample (M/P = 4/1, w/s = 0.14) [83].

Thus, solidification can be realized for a wide range of wastes:

— Chloride, nitrate, and radioactive nitrate-, nitrite-, and sulfate-containing solutions [84–87];
— Cs- and plutonium-contaminated ash [35,36,83,88,89];
— Reactive metals (Al, Mg, and U), which may corrode at high pH [90–95].

In addition, the $H_2$ radiolytic yield of MPC-based materials is 2–3 times less than that of PC-based materials because most of the mixing water participates in K-struvite formation. Gamma irradiation at a dose of 10 MGy has no notable effect on the mechanical performance and mineralogy of MPC mortars [96]. Bykov et al. [97] proposed a model of cement radiolysis in MPCs-based materials. The comparison of the radiation stability of PC and MPCs under γ-irradiation up to the absorbed dose of 100 MGy showed that the radiation-induced chemical decomposition of the materials in water was accompanied by the evolution of hydrogen. The evolution was retarded as the dose increased, where the limiting gas concentrations (~2.6 and ~0.7 L kg$^{-1}$ for PC and MPC, respectively) were reached at ~20 MGy, demonstrating that oxygen was entrapped by the constituents of the materials.

With the purpose to reduce the risk of radiolytic gas generation, as well as the corrosion of reactive metals in studies [98,99], MPCs were incorporated with fly ash (FA) and blast furnace slag (BFS). Gardner et al. [100] studied the behavior of blended MPCs at elevated temperatures to determine how waste packages behave when exposed to fire. The purpose

of research was formulated after fire and subsequent radionuclide release at the Waste Isolation Pilot Plant (WIPP) in the USA in 2014 [101]. Hardened pastes were exposed to a range of temperatures between 400 and 1200 °C to study the high-temperature behavior of FA/MKPC and GBFS/MKPC. At 400 °C, the dehydration of struvite-K ($MgKPO_4 \cdot 6H_2O$) was observed, leading to the loss of long-range crystallographic order. In the blended FA/MPC and GBFS/MPC binders exposed to temperatures of 1000 and 1200 °C, the formation of potassium aluminosilicate minerals (leucite and kalsilite), among other crystalline phases (hematite, spinel, and forsterite), was detected. The authors concluded that although the reactive phases assemblage and microstructure of the FA/MPC and GBFS/MPC binders were considerably altered at high temperatures, the binders formed stable products while retaining physical stability, with no evidence of spalling/cracking.

The possibility of solidifying borate-and nitrate-containing wastes was recently investigated by Lahalle et al. [102], Kononenko et al. [103], and Tao et al. [104]. Lahalle et al. [102] proposed a mechanism of retardation of MPC in the presence of borates. $B(OH)_3$ slows down the formation of hydrates in two ways: (i) by stabilizing in solution the cations that outbalance the negative charges of the polyborates formed at pH > 6 and (ii) through the precipitation of an amorphous mineral containing borate and orthophosphate. The first process proceeds in both diluted suspensions and pastes, whereas the second is specific to pastes. Kononenko et al. [103] used struvite-K ($KMgPO_4 \cdot 6H_2O$) and struvite ($NH_4MgPO_4 \cdot 6H_2O$)-based phosphate binders as a matrix for the solidification of liquid wastes. Authors simulated evaporator bottoms for a pressurized water reactor nuclear power plant (PWR NPP) with the following composition: $NaNO_3$—236.6 g dm$^{-3}$; $H_3BO_3$—168.2 g dm$^{-3}$; $NaOH$—189.6 g dm$^{-3}$; total salt content—509 g dm$^{-3}$ (37.3 wt%); pH—11.8; solution density—1.364 g cm$^{-3}$. The borates promoted struvite synthesis. The designed matrices contained up to 17–17.5 wt% salts, which was 1.7–2.5 times greater than that of the PC-based matrices. The volume of the struvite-based matrix was 1.6 times larger than the volume of the liquid waste from which it was obtained. With a Cs-selective nickel-potassium ferrocyanide sorbent or 10–20% MgO in excess of the reaction stoichiometry, the average rate of $^{137}$Cs leaching from the cementitious wasteform was less than $10^{-3}$ g·cm$^{-2}$·d$^{-1}$, with a mechanical strength over 5 MPa. Tao et al. [104] reported that incorporating simulated high-nitrate waste into MPCs changed the crystallization degree of struvite-K, where the microstructure changed from dense, plate-like, and prismatic crystals into loose, cluster-like crystals when the amount of nitrates exceeded 5%. Incorporating simulated high-nitrate waste into MPCs also retarded the hydration of the MKPC specimens and increased their porosity.

Vinokurov et al. [88] in 2009 studied MPCs as matrices for the solidification of simulated liquid alkaline high-level wastes containing actinides, as well as fission and corrosion products. These studies were continued by Lai et al. [81] who investigated the rapid immobilization of Cs and Sr in wastes from the PUREX process. The compressive strength of cemented wasteforms incorporating up to 50 wt% waste was 4.2 MPa and 13.2 MPa at an M/P ratio of 1 after 3 h and 1 d, respectively. The leaching rates of $Sr^{2+}$ and $Cs^+$ from the cemented forms were less than $10^{-7}$ g/m$^2$/d and $10^{-4}$ g/m$^2$/d, respectively. Zhenyu et al. [105] combined the benefits of ceramics and MPC materials. Particulate solidified ceramic forms with a composition of $Ca_{0.8}Ce_{0.1}TiSiO_5$ were first prepared by heating at 1300 °C for 2 h and then introduced into an MPC-based matrix. The obtained solidified forms demonstrated excellent mechanical properties, high-temperature stability, soaking resistance, and freeze–thaw resistance. The compressive strength of the samples decreased with increasing ceramic content, reaching 27.8 MPa with 50 wt% loading of the ceramic. However, the leaching rate of the simulated nucleus was found to be $1.86 \times 10^{-7}$ cm/d, which was less than that of the solidified ceramic form.

Pyo et al. [106] stated that radioactive concrete waste generated during the decommissioning of nuclear power plants can be effectively solidified using MPCs. The replacement of MPCs with 50% concrete waste even increased the compressive strength from 41 to 56 MPa. Moreover, the compressive strength remained >45 MPa after thermal-cycling and

water-immersion tests. The leaching indices of Cs, Co, and Sr, analyzed according to the ANS 16.1 procedure, were 11.45, 17.63, and 15.66, respectively.

However, based on a comprehensive analysis of solid waste-based MPCs, Zhang et al. [107] pointed out that the promotion of long-term, dynamic, and multi-dimensional research on MPC is an urgent task for the solid waste treatment of MPC.

MKPCs have also been described as "chemically bonded phosphate ceramics (CBPCs)" or by the trade name "Ceramicrete" and have been extensively developed and tested in the United States and Russia for conditioning various challenging nuclear wastes, including plutonium-contaminated ash, heavy metal and radium wastes, and [99]Tc-bearing wastes (using $SnCl_2$ as a reductant), as well as liquid Hanford vitrification wastes and Mayak salt wastes [34–38].

### 2.2.2. CPCs

CPCs comprise the calcium phosphates of diverse compositions or their blends with calcium salts (sulfate, carbonate, hydroxide, aluminate, calcium, etc.), magnesium orthophosphates, strontium, etc. [108–110]. The diverse combinations of calcium and phosphorus oxides (in the presence or absence of water) give a sufficiently large variety of different calcium phosphates; therefore, a wide range of raw materials is available for CPC production. The solubility in water, binding properties, and pH of the calcium phosphate cement are substantially influenced by the Ca/P ratio.

Hardened CPCs consist of stoichiometric or calcium-deficient hydroxyapatite. Their formation results from two reactions. A classic example of the first type of reaction is based on acid–base interactions; for example, the reaction of basic tetracalcium phosphate and acidic anhydrous dicalcium phosphate in an aqueous medium, leading to the formation of poorly crystallized hydroxyapatite (HA) [111]:

$$2Ca_4(PO_4)_2O + 2CaHPO_4 \rightarrow Ca_{10}(PO_4)_6(OH)_2$$

The second type of reaction involves the hydrolysis of metastable ortophosphate in an aqueous medium [110].

Hydroxyapatite resembles the structure of zeolites, characterized by presence of channels with diameters of 2.5 Å and 3–4,5 Å. Hydroxyapatite provides structural flexibility in the ion exchange with contaminant ions, often containing trivalent lanthanides and actinides, which can be replaced by Ca [22]. Hydroxyapatite also has low solubility, being 3–4 times less soluble than C-S-H and portlandite [30].

Recently, an efficient method for the consolidation of cobalt (Co(II))-adsorbed calcium hydroxyapatite was studied to design a simplified route for the decontamination of the coolant system of nuclear power plants and for the direct immobilization of the spent adsorbent [112]. Calcium hydroxyapatite nanopowder, produced by a wet precipitation method, was used as an adsorbent, resulting in a 94% removal of a Co(II) surrogate from simulated cooling water. The relative density after cold sintering was >97%; the obtained materials had a high compressive strength of 175 MPa. The normalized leaching rate of Co(II) was measured, as per the ASTM-C1285 standard, and found to be $2.5 \times 10^{-5}$ g/m²/d. The ANSI/ANS-16.1 test procedure was used to analyze the leachability of the sintered matrices, where the measured leaching index was 6.5.

### 2.3. Magnesium Silicate Hydrate Cements (M-S-H Cements)

M-S-H cements are based on the interaction of MgO or $Mg(OH)_2$ with amorphous silica, resulting in the formation of a M-S-H binder gel. This type of mineral matrix is relatively new among other cementitious materials for RW solidification but has already received scientific attention. Walling et al. [113] stated that Magnox sludge waste (a significant UK nuclear sector waste stream), consisting mainly of $Mg(OH)_2$, can be used as a primary constituent of M-S-H cement-based wasteforms, in combination with silica fume and an inorganic phosphate dispersant. Feasibility studies for the immobilization of $Cs^+/Sr^{2+}$ and Al by low-pH M-S-H cement have also been performed [114–116].

### 2.4. Alkali-Activated Materials

Invented more than 70 years ago, chemical, and particularly alkali, activation of glassy aluminosilicates—which is an approach for the non-fired or low-temperature production of inorganic binders from various natural and technogenic starting materials—has gained an ever-increasing appeal from the standpoints of theoretical research and industrial implementation, including the stabilization/solidification technique. This is largely due to the possibility of alkali-activated cements (AACs) achieving desirable properties, such as high fluidity, enhanced chemical resistance in aggressive environments, enhanced chemical tolerance to problematic and complex waste streams, potentially high waste loadings, and resilience against security of supply issues [18].

The general mechanism for the formation of hardened paste through alkali activation of glassy aluminosilicates consists of three different stages: (i) the destruction of aluminosilicate glass in an alkali medium, rupture of Si-O-Si and Al-O-Si bonds, and coagulation of transitional species, (ii) coagulation–condensation, and (iii) condensation–crystallization of calcium or sodium aluminosilicate hydrogel as a major reaction product [117,118]. The distinguishing feature of AAC is the greater number of influencing factors than those in PC-based systems. Generally, the formation process, structure, and properties of AACs depend on many factors, including the following: (1) precursor factors, such as the shape and size of the particles, crystal/vitreous phase ratio, and chemical composition (e.g., reactive $SiO_2/Al_2O_3$ and CaO content); (2) alkali activator factors, such as type (MOH, $M_2O \cdot rSiO_2$ ($SiO_2/Na_2O$), and $NaAlO_2$), molarity, pH, and addition methods (e.g., dry form and solution form); as well as (3) processing factors, such as grinding and mixing methods and the curing regime (e.g., temperature, humidity, and time) [28,119–124]. By varying the controlling parameters, it is possible to design AAC-based cementitious materials with pre-determined reaction products and physical performance, enabling efficient RW encapsulation, as well as the sequestration of specific contaminants and wastes. As alkali activation allows the use of precursors with a wide range of chemical compositions in terms of the percentage of reactive Ca, Si, and Al, the reactive phases assemblage of hardened AACs varies widely. Fast setting, high strength, low porosity, and high chemical and heat resistance are typical for the appropriate formulations of AACs. The range of potential starting materials has changed and expanded continuously throughout the history of AACs [28,119,125–128]. As regards the sources of AACs used as matrices for the immobilization of RW; granulated BFS, FA, and MK; and their combinations, they are now the basic precursors, whereas sodium and potassium hydroxides and silicates are normally used as alkali reactants.

Historically, studies in the field of AACs as cementitious matrices for RW began in the early 1990s, with initial studies on BFS-based AACs [39,129]. The alkali activation of high-Ca precursors, including BFS, results in the formation of a tobermorite-like aluminum-substituted calcium silicate hydrogel C-(A)-S-H [130,131].

Many studies [18,19,23,24,26,28–32,132–134] have demonstrated the efficiency of BFS-based AACs for the solidification of wastes containing heavy metals (such as $Zn^{2+}$, $Pb^{2+}$, $Cd^{2+}$ and $Cr^{6+}$, $Hg^{2+}$, etc.) and radionuclides ($Cs^+$, $Sr^{2+}$), as well as sodium-borate-containing liquid wastes, ion-exchange resins, etc.

Studies on the mechanisms of action of the BFS-based AACs with $Cs^+$ and $Sr^+$ have been conducted in the last few years by several researchers [135–137]. Vandevenne et al. [135] evaluated the mechanism of immobilization of $Cs^+$ and $Sr^+$ (0.5–2% wt%) by 6 M NaOH-activated BFS-based AAC. The authors reported that $Cs^+$ was almost fully incorporated into the mineral matrix, whereas $Sr^{2+}$ mainly precipitated as $Sr(OH)_2$ throughout the AAC-hardened paste. Huang et al. [136] reported that the addition of sodium hexametaphosphate to sodium silicate/sodium hydroxide-activated BFS1 paste enhanced the chemical binding of $Sr^{2+}$ ions via hydroxyapatite formation and $Sr^{2+}$ substitution. Microwave irradiation further increased the mechanical performance of the hardened pastes and inhibited the leaching of $Sr^{2+}$ ions from the matrices by strengthening hydration reactions and $Sr^{2+}$ encapsulation. According to Komljenovic et al. [137], introducing 2% and 5%

Cs into sodium silicate BFS-paste increased the early strength of the hardened paste, with no noticeable effect on the composition of the binder gel.

Since the end of the 70s of the last century, increasing attention has been paid to a subclass of AACs now termed "geopolymers", which are based on low-Ca or Ca-free precursors, such as class F FA and MK. The major reaction product of alkali-activated FA or MK is a three-dimensional, cross-linked, and structurally disordered sodium aluminosilicate hydrate gel, N-A-S-H (Figure 7). The binding gel comprises Si and Al in tetrahedral coordination, connected by oxygen atoms in a pseudo-zeolitic framework structure. Si exists in $Q^4$(mAl) environments ($1 \leq m \leq 4$, depending on the Al/Si ratio of the gel). The negative charge arising from $Al^{3+}$ in tetrahedral (four-fold) coordination is charge-balanced by the alkali cations provided by the activating solution, commonly Na or K. The secondary reaction products are zeolites, such as hydrosodalite, zeolite P, Na-chabazite, zeolite Y, and faujasite [118] (Figures 7 and 8). Such a reaction product assemblage is favorable for the chemical binding of many contaminants, providing high immobilization potential, accompanied by high physical–mechanical performance of the hardened wasteforms. Thus, the geopolymers have higher efficiency than AACs based on high-Ca precursors, fueling increasing studies in this field started by Davidovits et al. [40,41] at the end of 1990s. Thus far, the binding efficiency of geopolymers for 37 elements, including Sr, Cs, Pb, Cr, and Zn, has been proven [24,28,138,139]. However, the encapsulation of reactive metals and oils by geopolymers and the radiolysis of water in the binder gel under gamma irradiation require further investigation and new approaches [7,140–144]. It is worth noting that geopolymers have been intensively and increasingly studied in recent decades for RW solidification (Figure 9) [107].

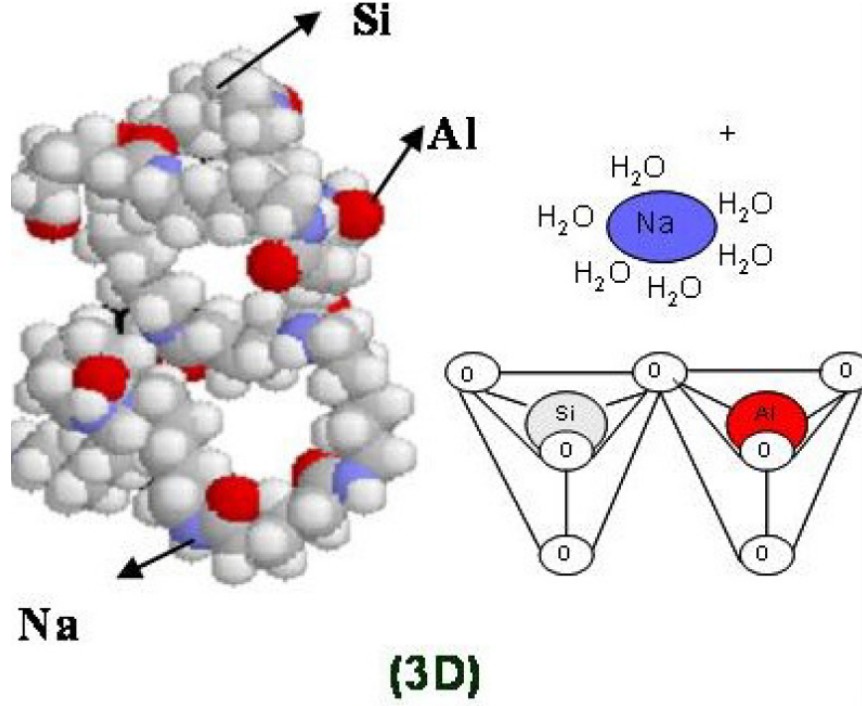

**Figure 7.** N-A-S-H gel structure [118].

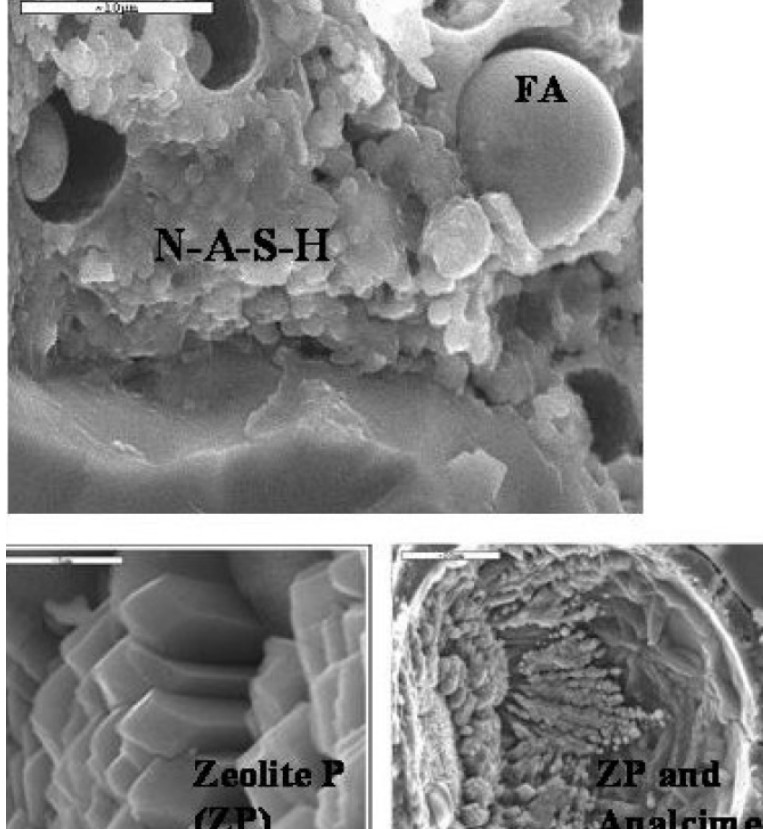

**Figure 8.** SEM micrographs of N-A-S-H and zeolites [118].

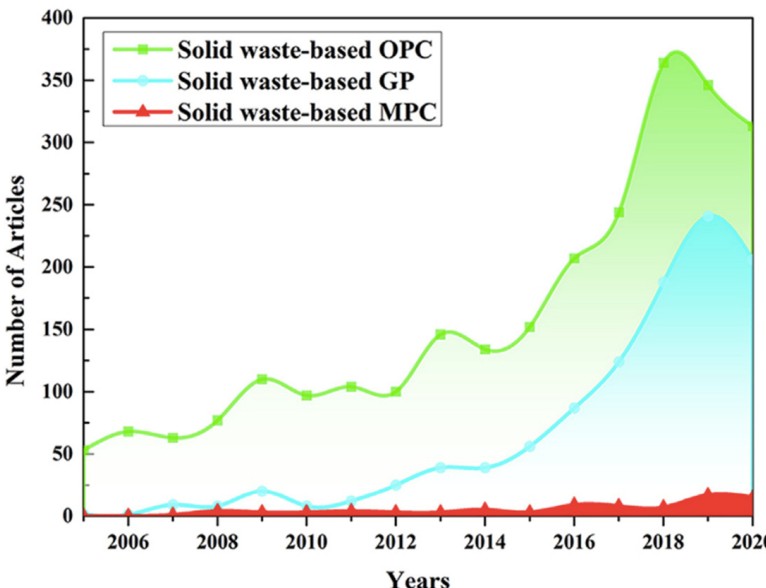

**Figure 9.** Number of OPC-, GP-, and MPC-related references in the past 15 years [107].

The immobilization of $Cs^+$ and $Sr^+$ remains the subject of ongoing research. The results presented by Walkley et al. [139] contribute to further understanding the mechanism of Sr and Ca immobilization in MK-based geopolymers. The incorporation of alkaline earth cations resulted in a minor decrease in the Si/Al ratio of the (N,K)-A-S-H gel; no other changes were found for pastes hardened at 20 °C; however, in those cured at 80 °C,

the incorporation of Sr appeared to promote the formation of zeolite A over the faujasite zeolite phases. According to El Alouani et al. [145], the kinetics of $Cs^+$ adsorption by AACs followed pseudo-first-order and pseudo-second-order kinetic models, indicating that both the physisorption and chemisorption mechanisms controlled the adsorption process. Many studies have stated different ways for effective chemical immobilization of Cs and Sr by geopolymers, such as: (i) in the form of clinoptilolite incorporated with Cs and Sr, Sr-loaded zeolite A [146,147], (ii) Sr-loaded titanate spent adsorbents [148,149], (ii) Cs waste as an activating solution [150–152]; (iii) Cs and Sr hydroxides [153], and (iv) sewage sludge ash contaminated with radiocesium [154]. Lin et al. [155] reported that MK-based AAC safely and effectively solidified IERs (up to12 wt%) for immobilizing both $Cs^+$ and $Sr^{2+}$. Tan et al. [156] also found that the MK-based AAC binder exhibited better leaching resistance than the PC binder in deionized water, solution of $H_2SO_4$, $MgSO_4$, and acetic acid buffer. The compressive strength of MK-based AAC declined to a lesser extent after freeze–thaw cycles and high-temperature tests than that of PC.

Based on the presented data, Arbel-Haddad et al. [157] concluded that designing GP formulations to provide a higher amount of zeolite F is reasonable for the production of matrices to immobilize Cs, because Cs is mainly bound by zeolite F, rather than by other reaction phases of low-Si MK geopolymers.

Curing at elevated temperatures and high-temperature sintering of the hardened products, leading to the formation of different crystalline phases in AAC production, is one approach for improving $Cs^+$ immobilization. Fu et al. [158] observed the formation of analcime and pollucite in Na- and Cs-rich MK-based systems cured at 170 °C under hydrothermal conditions.

Chaerun et al. [159] found that K-based MK-GP incorporated with a chabazite adsorbent was more effective than Na-based AAM or PC for the immobilization of Cs. The crystallographically disordered nature of K-AAM and its pH were the main contributors to K-ion migration and the structural change of aluminosilicate rings in chabazite, thus resulting in the formation of K-type chabazite with amorphous properties similar to those of K-AAM. Due to the similar ionic radii and retention selectivities of Cs and K, both can be confined during the reconstruction of the aluminosilicate and are crystallized into pollucite during the AAM fabrication process. Jain et al. [160] investigated the effect of the Cs content on the reaction products and pore structure of FA-based geopolymers. A higher Cs loading ($\geq 8$ wt.%) facilitated in situ pollucite crystallization within the FA-GP matrix (cured at 90 °C for 7 d) and significantly enhanced Cs immobilization (leachability index of 11.5–14.5).

Ahn et al. [161] investigated MK-based geopolymers for the solidification of sulfate-rich HyBRID sludge waste, consisting of cristobalite ($SiO_2$) and barite ($BaSO_4$) as major components. The K-based geopolymer had a higher mechanical strength (up to 14.3 MPa) than the Na-based geopolymer, and could also solidify more HyBRID sludge waste, thereby increasing the waste loading to 53.8 wt%. The pure geopolymer with the HyBRID sludge waste exhibited good mechanical stability at a Si/Al ratio of 1.8. However, the highest compressive strength was achieved for the geopolymer prepared with 40 wt.% HyBRID sludge waste at a Si:Al ratio of 1.6. Authors attributed these differences to the consumption of water and additional Si sources.

Kim et al. [162] investigated the potential of simulated borate waste (sodium tetraborate decahydrate ($Na_2B_4O_7 \cdot 10H_2O$), sodium nitrate ($NaNO_3$), potassium nitrate ($KNO_3$), calcium nitrate ($Ca(NO_3)_2 \cdot 4H_2O$), zinc nitrate ($Zn(NO_3)_2 \cdot 6H_2O$), and magnesium nitrate ($Mg(NO_3)_2 \cdot 6H_2O$)) as raw materials for producing MK-based geopolymers, all of which had higher compressive strengths than the PC-based cementitious wasteforms. The K-geopolymers (40 MPa) had a higher 7 d compressive strength than the Na-geopolymers (24 MPa). However, the compressive strength tended to increase in proportion to the Si/(Al+B) ratio (1.3–1.5), irrespective of the type of alkaline cation. This variation was attributed to the viscosity of the activator used for geopolymer formation, the atomic size

of the alkaline cations, and the increase in the Si content. However, as shown in another study [163], the addition of borax increased the reactivity and geopolymer polycondensation.

He et al. [164] compared the temperature-dependence (from 25 °C to 60 °C) and environment-dependence of the kinetics of $Sr^{2+}$ and $Cs^+$ leaching during long-term leaching tests and reported a low leaching rate and relatively high mechanical properties of Na geopolymers. A high storage temperature and salt medium accelerated the leaching of $Sr^{2+}$ and $Cs^+$ from the matrix by enhancing the driving force of the leaching process and the corrosion effect. Compared with the change in temperature, the leaching of $Sr^{2+}$ and $Cs^+$ is more sensitive to changes in the leaching medium, indicating that the corrosion effect of the salt medium plays a more important role in accelerating the leaching of radioactive elements and degradation of the immobilizing matrix.

The performance of AACs as cementitious materials can be effectively manipulated by using mixed precursors. Combining Ca-rich and Ca-free starting materials in alkali-activated systems produced chemically and structurally different binder gels, such as calcium-containing (C-(A)-S-H) and calcium-free (N-A-S-H) [165,166] gels, as well as mixed (C-N-(A)-S-H) gels. C-N-(A)-S-H represents mechanically strong gels consisting of crystalline tobermorite-like and amorphous cross-linked products with a relatively high content of silica in $Q^1$, $Q^2$, and $Q^3$ sites, which leads to the densification of the binder–gel microstructure [167–169]. Densification positively affects the physical–mechanical properties and immobilizing properties of the matrixes [170–178].

AACs based on both high-Ca and Ca-free precursors demonstrated good effects in the solidification of IERs with loadings up to 60% (by volume) (Table 3).

**Table 3.** The formulations and properties of AACs incorporated with IERs.

| Precursor | Alkali Reactant | Ion-Exchange Resins | Details | Ref. |
|---|---|---|---|---|
| Ground granulated BFS—100% (wt%) | $Na_2SiO_3 \cdot 9H_2O$ (NSH$_9$)+ sodium hydroxide. (NaOH) (5–7% by $Na_2O$) | Loading of cationic borate IERs 35% by volume (pH 8.5–10.5) | 28 d compressive strength up to 7.3 MPa | [179] |
| Ground granulated BFS—31–48% (wt%), wollastonite—6–8% (wt%) | 9–12 M NaOH ($SiO_2/Na_2O$ = 0.8 and $SiO_2/Al_2O_3$ = 50) | Loading of wet IERs 45%, dry IERs 22% (wt)) | 28 d compressive strength up to 22 MPa. ($SiO_2/Na_2O$ = 0.8 and $SiO_2/Al_2O_3$ = 50) | [180] |
| MK 90%, Ground granulated BFS—31–48% (wt%) | Sodium silicate/sodium hydroxide 58% (wt%) | Loading of wet IERs 12% (wt) | 28 d compressive strength 13.9 MPa | [155] |
| FA 56% (wt%) | $NaOH/Na_2SiO_3$ 28% (wt%) | 10% (wt) | 28 d compressive strength 13.9 MPa | [181] |
| Ground granulated BFS—35–40% (wt%), bentonite 2.5–8%, $Ca(OH)_2$—4–5%, or OPC—8–10% | $Na_2SiO_3 \cdot 5H_2O$—6% or $Na_2CO_3$—1.5–3% | Loading of cationic and anionic IERs 60% by volume | 28 d compressive strength up to 18 MPa | [182] |

In Czech Republic, the waste conditioning using geopolymers has been carried out at the Dukovany NPP by the external supplier AMEC Nuclear Slovakia. By the end of 2012, the Dukovany NPP Units 1 and 2 extracted, conditioned, and disposed of more than 200 t of sludge and IX resins [42].

In Slovak Republic, AllDeco Ltd has developed a proprietary geopolymer matrix (called SIAL) for embedding various intermediate-level wastes resulting from Slovak

power reactors [163]. Some of the materials encapsulated in geopolymer matrices are bottom sludges from the long-term storage of spent nuclear fuel elements, sludges from the sedimentation tank of a reactor, and several other sludges. Some of the sludges are formed from an emulsified mixture of organic compounds from the cooling media and contain a large amount of calcium and magnesium hydrocarbonates. The activity of the $^{137}$Cs in the sludges is ~105–108 Bq/L. Once these sludges were solidified in the geopolymer matrix and placed in 60 L drums, the surface dosage on the drums was 10–20 mGy/h. The D value for the 137Cs in samples taken from the drum was >8 for the ANSI 16.1 test and the compressive strength was 25 MPa. About 20 wt% (on a dry basis) of waste was encapsulated. Organic ion exchange resins on their own and in mixtures of sludges were also encapsulated in geopolymer matrices. It was possible to encapsulate ~20 wt% (on a dry basis) for geopolymers compared to 10 wt% for OPC. These were placed in 200 L drums. The dosages on the drum surfaces were 130–600 μGy/h and the D value (leachability index) for $^{137}$Cs was >9 on cut samples from the drums. All the drums used were made from stainless steel. The SIAL matrix (geopolymer) has been accepted by the Slovak Nuclear Authority (UJDSR) and the Czech Nuclear Authority (SUJB) for placement in their respective repositories. AllDeco Ltd emplaced these drums in the Slovak repository in 2003.

In Australia, ANSTO geopolymers derived from metakaolin and alkaline silicate solutions with nominal Na/Al and Si/Al molar ratios of 1 and 2 were studied for the stabilization of $^{137}$Cs and $^{90}$Sr with Cs inhabited the amorphous phase, whereas Sr was incorporated only partly, being preferentially partitioned to crystalline $SrCO_3$ [43,44].

In the USA, geopolymers with Si to Al ratios of 1 to 1 and 2 to 1 were investigated for the stabilization of hazardous Resource Conservation and Recovery Act (RCRA) metals, such as Ni, Se, Ba, Hg, Cd, Cr, and Pb [44]. Special geopolymer formulations, marketed under the name DuraLith, have been patented for stabilization of 129I and 99Tc at Hanford Waste Treatment Plant [44]. The DuraLith geopolymer is composed of three components: an activator, a binder, and an enhancer [45]. DuraLith is an alkali-activated geopolymer waste form developed by the Vitreous State Laboratory at The Catholic University of America for encapsulating liquid radioactive waste. A DuraLith waste form developed for treating Hanford secondary waste liquids is prepared by the alkali-activation of a mixture of ground blast furnace slag and metakaolin with sand used as a filler material. The DuraLith geopolymers demonstrates compressive strength above 27 MPa, and ANSI/ANS 16.1 Leachability indexes for Tc as high as 9. Savannah River Site has used the FA-based geopolymer. The Class F FA resulted from the burning of harder, older anthracite and bituminous coal and is pozzolanic in nature, containing less than 7% CaO. Adding a chemical activator, such as sodium silicate (water glass) to a Class F ash can form a geopolymer. The wasteform geopolymer recipe contained in wt.%: waste granules 47.4, Class F ash 12.8, $Na_2O$ $2SiO_2$ 44.1, NaOH (50 wt%) 12.5, and water 8.1.

## 3. Conclusions

Modern mineral matrices for nuclear waste immobilization now include a wide class of cementitious materials with various chemical–mineralogical compositions, high encapsulation capacities, and technological and engineering performance, comprising not only traditional Portland cements but also non-Portland clinker inorganic binders. Consistent development of research in this field has extended the theoretical basis, potential, and versatility of cementation technology. Based on the review and analysis of trends and achievements in the immobilization of nuclear wastes using CACs, CSACs, M-S-H, phosphate, and AACs, the following conclusions can be drawn:

1. Scientific and practical interest in the use of alternative cementitious materials for nuclear waste treatment and conditioning has only increased in recent years.
2. The appropriate formulations of alternative cements combine the high ion-exchange capacity and high physical–mechanical properties of hardened pastes, such as quick setting, high strength, high temperature, and chemical resistance. The design of cemen-

titious materials with "targeted" reactive phase assemblage and excellent physical–mechanical performance is achievable by: (i) varying a range of factors that govern the properties of binder systems; (ii) introducing supplementary cementitious materials into CACs, CSACs, and phosphate cements.

3. Radionuclides (Cs, Sr, etc.), borate- and nitrate-containing wastes, oils, IERs, solid wastes, etc., remain the subject of a great number of studies, most of which highlight the superiority of alternative cements as solidifying matrices compared to PC in terms of effective physical encapsulation and chemical binding of RW, the waste loading rate, and durability of the cementitious wasteforms.

4. New insights into the mechanism of action of Cs, Sr, B, I, etc., as well as the resultant reaction products, have been proposed by several researchers. Presented results proved high efficiency of both chemical binding and physical encapsulation capacity of alternative cementitious materials.

5. The cementation of RW as a "raw material" for cements is a new perspective trend of cementation technique, demonstrating good results. Thus, Cs waste as an activating solution and simulated borate waste were used in recent studies for producing geopolymers.

6. Several studies have demonstrated that phosphate cements and AACs are effective for high-level waste immobilization. MPCs provided good results in rapid immobilization of Cs and Sr in wastes from the PUREX process.

Further studies in the development of alternative-cements-based mineral matrices for the solidification of various types of RW, as well as research into reaction mechanisms of matrices and contaminants, long-term immobilization and durable performance of the wasteforms, the behavior of the cemented wastes in emergency situations, new analytical techniques, and predictive computational modeling of cementitious wasteforms, will contribute to the further sustainable development of RW management systems.

**Funding:** This research received no external funding.

**Institutional Review Board Statement:** Not applicable.

**Informed Consent Statement:** Not applicable.

**Data Availability Statement:** Not applicable.

**Conflicts of Interest:** The authors declare no conflict of interest.

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
