# Peer review of "Recent Advances in Alternative Cementitious Materials for Nuclear Waste Immobilization: A Review"

_sustainability, doi:10.3390/su15010689_

Round 1

Reviewer 1 Report

The manuscript in a straightforward manner describes the defined problem. It contains all the necessary elements of the review paper. The number of supporting references is sufficient.

The following improvements should be provided:

a)      The more comprehensive introduction (one paragraph) about general problems related to the radioactive waste should be provided. This will make the reader more familiar with the problem. I strongly recommend adding references to the tools and methods for burnup calculations, which in principle serve to estimate isotopic composition, activity, and generated heat of the spent nuclear fuel, which in turn is  crucial for the choice of storage options. Recently, some papers on trajectory period folding method for burnup calculations and application of radiochemical measurements of PWR spent fuel for the validation of burnup codes, were published in the MDPI Energies journal.

b)     The manuscript lacks the section ”Discussion” – please add.

c)      In “Conclusions” please develop points 4,5,6. More insight is necessary.

d)     Please add a more comprehensive description of the future promising research in the field (in conclusions or discussion section).

Reviewer 2 Report

Review of N. Rakhimova "Recent Advances in Alternative Cementitious Materials for Nuclear Waste Immobilization: A Review // Sustainability 2022, 14, x" is important and usefull scientific report. It focuses on recent achievements in the development of various cements as cementitious matrices suggested for nuclear waste stabilization / solidification.

It is recommended for publication with minor correction.

Some comments and questions are as followed:

1) In the section about geopolimers, including the Table 1, there no any data and information on the works and research of the famous french scientist prof. J. Davidovits. He studied the problem in relation to radioactivw waste also.

2) Leaching rate (Figure 4) for elements (radionuclides) is usually determined in units: gramms from cm2 per day. 
